# Topical Propranolol Improves Epistaxis Control in Hereditary Hemorrhagic Telangiectasia (HHT): A Randomized Double-Blind Placebo-Controlled Trial

**DOI:** 10.3390/jcm9103130

**Published:** 2020-09-28

**Authors:** Meir Mei-Zahav, Yulia Gendler, Elchanan Bruckheimer, Dario Prais, Einat Birk, Muhamad Watad, Neta Goldschmidt, Ethan Soudry

**Affiliations:** 1Pulmonary Institute, Schneider Children’s Medical Center of Israel, Petah Tikva 49202, Israel; yulia.gendler@gmail.com (Y.G.); prais@tauex.tau.ac.il (D.P.); 2Sackler Faculty of Medicine, Tel Aviv University, Tel Aviv 69978, Israel; elrealview@gmail.com (E.B.); EBirk@clalit.org.il (E.B.); Watadmohammed@gmail.com (M.W.); ethansoudry@gmail.com (E.S.); 3The National HHT Center, Pulmonary Institute, Schneider CMCI, 14 Kaplan St., Petach Tikva 49202, Israel; 4The Department of Nursing, Ariel University, Ariel 40700, Israel; 5Cardiology Department, Schneider Children’s Medical Center of Israel, Petah Tikva 49202, Israel; 6Department of Hematology, Hadassah-Hebrew University Medical Center, Jerusalem 91120, Israel; neta@hadassah.org.il; 7Department of Otolaryngology Head and Neck Surgery, Rabin Medical Center, Petah Tikva 49202, Israel

**Keywords:** hereditary hemorrhagic telangiectasia, epistaxis, propranolol gel, epistaxis severity score, nasal endoscopy, antiangiogenic properties

## Abstract

Epistaxis is a common debilitating manifestation in hereditary hemorrhagic telangiectasia (HHT), due to mucocutaneous telangiectases. The epistaxis can be difficult to control despite available treatments. Dysregulated angiogenesis has been shown to be associated with telangiectases formation. Topical propranolol has demonstrated antiangiogenic properties. We performed a two-phase study, i.e., a double-blind placebo-controlled phase, followed by an open-label phase. The aim of the study was assessment of safety and efficacy of nasal propranolol gel in HHT-related epistaxis. Twenty participants with moderate-severe HHT-related epistaxis were randomized to eight weeks of propranolol gel 1.5%, or placebo 0.5 cc, applied to each nostril twice daily; and continued propranolol for eight weeks in an open-label study. For the propranolol group, the epistaxis severity score (ESS) improved significantly (−2.03 ± 1.7 as compared with −0.35 ± 0.68 for the placebo group, *p* = 0.009); hemoglobin levels improved significantly (10.5 ± 2.6 to 11.4 ± 2.02 g/dL, *p* = 0.009); and intravenous iron and blood transfusion requirement decreased. The change in nasal endoscopy findings was not significant. During the open-label period, the ESS score improved significantly in the former placebo group (−1.99 ± 1.41, *p* = 0.005). The most common adverse event was nasal mucosa burning sensation. No cardiovascular events were reported. Our results suggest that topical propranolol gel is safe and effective in HHT-related epistaxis.

## 1. Introduction

Hereditary hemorrhagic telangiectasia (HHT) (ORPHA774, HHT 1 OMIM# 187300 and HHT 2 OMIM# 600376) is an autosomal dominant vascular disorder that leads to abnormally dilated blood vessels and arteriovenous malformations. Telangiectases tend to rupture and result in recurrent gastrointestinal bleeding and spontaneous epistaxis, the latter being the most common clinical presentation of patients with HHT [1]. Severely affected individuals may have gushing bleeds several times a day, with consequent iron deficiency anemia and transfusion dependency [1]. Severe epistaxis is associated with significant impairment in daily life activity and decreased quality of life (QOL) [2,3,4], and is often resistant to standard measures [5].

Although the precise mechanism of arteriovenous malformations and telangiectases formation in HHT is unknown, dysregulation of angiogenesis has been shown to result in an unbalanced generation of abnormal blood vessels [6]. Several antiangiogenic therapeutic agents have been proposed. Bevacizumab, the most frequently used, has demonstrated limited success as a topical treatment of epistaxis [7,8], and significant side effects with systemic use [9].

The non-selective beta-blocker propranolol is commonly used as a topical and systemic treatment for infantile hemangioma [10], with proven anti-angiogenic effects [11]. We previously reported the efficacy of topical propranolol in an open-label study with six participants [12]. That report led us to conduct the current trial to evaluate the safety and efficacy of topical propranolol gel in the treatment of moderate to severe HHT-related epistaxis.

## 2. Experimental Section

This study was comprised of a double-blind placebo-controlled phase, with continuation to an open-label phase (Figure 1). Patients with HHT having refractory moderate to severe epistaxis were recruited from the Israeli National HHT Center at Schneider Children’s Medical Center of Israel and Beilinson Medical Center. Study inclusion and exclusion criteria are summarized in Table 1. Patients with concomitant significant gastrointestinal bleeding were referred for systemic treatment and were not included in the study. 

The study was approved by the local ethics committee (RMC-0191-15) and by the Israel Ministry of Health (MOH, Clinical Trial Registration 20173382, www.health.gov.il/clinicaltrials). All the participants signed an inform consent.

Propranolol gel was prepared with propranolol HCl 1.5% in an isotonic solution and preserved hydroxyethyl cellulose 2% gel. The placebo gel was prepared with preserved hydroxyethyl cellulose 2% gel only. Patients were instructed to apply 0.5 cc propranolol or a placebo gel via a supplied 1 mL syringe, to the nasal mucosa of each nostril, twice daily.

Following screening, the patients were randomly assigned, at a 1:1 ratio, to be treated with propranolol gel or a placebo gel for a period of 8 weeks (the double-blind phase). Randomization was performed by the pharmacy, and the treatment team was blinded to the process. Following the double-blind phase, all the patients were offered to continue an open-label phase for an additional 8 weeks.

The study participants were examined at screening and randomization, and at the end of each treatment phase (the 8th and 16th weeks). Each visit included a recent medical history and physical examination; monitoring of side-effects; measurements of heart rate (HR), blood pressure (BP), blood hemoglobin (HB), and iron and ferritin levels; a rhinology examination; assessment of epistaxis severity according to the epistaxis severity score (ESS); and quality of life (QOL) according to the 12-Item Short Form Health Survey (SF-12) questionnaire. An electrocardiogram was performed at screening and repeated at the primary investigator’s discretion.

Nasal endoscopy was performed at screening, and at the 8th and 16th weeks, by a highly trained rhinologist (ES), who was blinded to treatment allocation.

Patients’ nasal cavities were decongested with lidocaine 1.5% and phenylephrine 1% spray prior to endoscopic examination with a zero-degree 4 mm endoscope connected to a high definition camera and monitor (Storz). Endoscopies were recorded and representative photographs of the nasal cavities were captured in a de-identified manner. Thus, all patient images were subsequently graded in an anonymized fashion at the conclusion of the study. Nasal involvement with disease at recruitment was graded as follows: mild-few punctate telangiectases, moderate-multiple telangiectases/large arteriovenous malformations involving the anterior nasal septum, and severe-diffuse involvement of the nasal mucosa with telangiectases. During follow up, endoscopies were defined as improved versus no change or worsened.

A telephone interview was performed at each mid-treatment period (4th and 12th week) and 4 weeks after the open-label phase (20th week) to assess efficacy and safety.

Participants were instructed at recruitment to fill out a daily epistaxis diary indicating the severity, frequency, and duration of the epistaxis episodes. Severity was recorded as mild-drops of blood (1), moderate- mild bleed and clots (2), or severe gushing or major clots (3).

Bleeding frequency was recorded as the number of episodes per day. The total minutes of epistaxis per day was recorded as the daily epistaxis duration. Participants were asked to mark in the diary when the medication was applied. Compliance was measured using the diary and by counting the empty syringes returned at every visit. Participants were instructed to measure their BP and HR weekly by their local health provider and to document these results in their diary. Participants were also asked to document any local or systemic symptoms and report them to the investigators.

Indications to terminate the study were any of the following criteria: a systolic BP drop to less than 80 mmHg or a drop of ≥20% from baseline systolic BP, a drop of HR to less than 50/min, any signs of heart block on the electrocardiogram, grade > 2 (CTCAE [15]) local or systemic side effects such as local irritation or an allergic reaction to the medication, and patient’s or physician’s request.

The primary outcome was the difference in ESS drop between both groups. ESS was first calculated at randomization and was related to the prior 8 weeks, and then at the end of the double-blind period. The change in ESS from randomization to the end of the first 8-week period was compared between the propranolol and control groups. The secondary outcome measures were changes in blood hemoglobin (Hb) levels and intravenous iron, packed cell transfusion (PC) requirements, a change in the intensity of telangiectases in the nasal mucosa as documented by one otolaryngology surgeon (ES), and the change in QOL. In the open-label period the secondary outcome was ESS change from the beginning and the end of the open-label phase. 

### Statistical Analysis

The sample size was calculated from a pilot study [12], in which ESS improved by 2.85 ± 1.75 following the use of propranolol gel. We anticipated a similar improvement in the current study. We assumed an improvement in ESS of 0.5 in the placebo group. Power analysis calculation (α of 0.05 and 1-β of 0.8) required 9 patients in each arm. Assuming that two participants would drop out of the study per group, 11 participants were planned to be recruited in each group (for a total of 22 participants in the study). 

Data were analyzed using SPSS, version 25. Armonk (NY, USA) (SPSS Inc., Chicago, IL, USA). Demographic factors and clinical outcomes were summarized with percentage breakdown. For comparing outcomes between the treatment groups, an independent samples *t*-test was used, with a normal distribution assumed. Otherwise, Mann–Whitney or Wilcoxon rank-sum tests were used. Changes in ESS and clinical parameters between two visits were analyzed using a paired *t*-test. To analyze the improvement in nasal rhinology scoring between the two groups (improved/not improved), the McNemar test was used. Normality assessment was performed using a Shapiro–Wilk test. Normal distribution was assumed when *p* > 0.05.

## 3. Results

### 3.1. Study Participants

Twenty-three patients were recruited between April 2018 and March 2019, of whom, 18 (78%) were female. The mean age at study entry was 54.6 ± 10.8 years (range 35–74 years).

Ten participants were randomized to the propranolol group, and 13 to the placebo group. Three participants from the placebo group withdrew from the study, two because of low compliance and one after the development of acute otitis media. None of the participants in the propranolol group withdrew.

Twenty participants completed the double-blind phase, 10 participants in each study group. Their baseline characteristics are presented in Table 2. Differences in the baseline ESS, and in demographic and clinical parameters between the groups were not statistically significant.

Compliance was high for both groups (propranolol 90.2 ± 10.3% vs. placebo 94.2 ± 8.1%, *p* = 0.371). 

### 3.2. Primary Outcome

During the first phase of the study, the ESS showed a significant improvement in the propranolol group (from a mean of 6.50 ± 1.84 to 4.47 ± 1.75, *p* = 0.004) and no change in the placebo group (5.68 ± 1.8 to 5.33 ± 2.1, *p* = 0.133, Table 3). The change in ESS was −2.03 ± 1.7 in the propranolol group as compared with −0.35 ± 0.68 in the placebo group, *p* = 0.009 (Figure 2, Table 4).

Box and whisker plots of the decreases in ESS in the placebo and propranolol groups during the double-blind phase of the study. The black horizontal lines represent the median values of decrease and the range. The dots represent the mean and the boxes represent the interquartile range of decrease (25% to 75%). ESS change of 0.71 was suggested to be the MID (minimal important difference) [16].

### 3.3. Secondary Outcomes

Significant improvement in Hb level was observed in the propranolol group but not in the placebo group (Table 3). PC transfusion requirement decreased significantly in the propranolol group. Changes were not observed in these parameters in the placebo group. The change in intravenous iron units required was not statistically significant in both groups. QOL improved significantly in both groups.

We also compared the change in the secondary outcomes during the study period between the placebo and the propranolol groups (Table 4). The propranolol group had a significant decrease in PC requirement as compared with the placebo group. There was no significant change in the other secondary outcome measures.

### 3.4. Nasal Endoscopy

Four patients in the placebo group improved (40%), of whom three were grade I at study entry. Improvement was observed in seven patients (70%) in the propranolol group, of whom five were grade II/III at study entry. One patient in the propranolol group worsened. The difference in improvement between the groups at the end of the placebo-controlled phase was not significant (*p* = 0.18). Among patients with higher grading (II and III) of nasal involvement, improvement was observed in 71.4% of the propranolol group as compared with 16.7% of the placebo group (*p* = 0.078).

### 3.5. Side Effects

Hypotension or bradycardia were not observed among participants in the placebo and propranolol groups. Mean BP measures were similar between the groups (systolic BP 108.5 ± 8.1 mmHg vs. 111.3 ± 5.95 mmHg, *p* = 0.402 and diastolic BP 60.2 ± 7.6 mmHg vs. 67.5 ± 12.6 mmHg, *p* = 0.137), as was the mean heart rate (73.6 ± 9.7 beats per minute vs. 74.4 ± 13.45, *p* = 0.876).

The most important side effect observed was a burning sensation in the nasal mucosa and the pharynx (Table 5). Five patients in the propranolol group and two in the placebo group had mild and transient burning sensations that resolved during treatment. Four patients in the propranolol and none in the placebo group complained of a substantial burning sensation. Overall, 9 patients in the propranolol group and 2 in the placebo group had some burning sensation (*p* = 0.005). All the patients had a thorough otolaryngologic examination and no visible mucosal injuries were observed. Rhinorrhea was observed in three patients in the propranolol group and in none in the placebo group (*p* = 0.06). Nasal dryness was observed in one participant in the placebo group.

One patient in the propranolol group had acute otitis media. One patient in the placebo group withdrew the study because of otitis media which led to her hospitalization. The patients both recovered. No other side effects were observed.

### 3.6. Results of the Open-Label Phase of the Study

Fifteen participants completed the open-label phase, seven in the propranolol group and eight in the placebo group. Two patients in the placebo group withdrew from the study at this phase, as they were awaiting a Young’s procedure prior to the study and decided to proceed with this process. Three patients in the propranolol group withdrew from the open-label label phase because of a burning sensation; two of them also had rhinorrhea.

During the open-label phase, the former placebo group, now receiving propranolol gel, showed significant improvement in the ESS score, with a mean drop of 1.99 ± 1.41 from 4.86 ± 1.85 to 2.87 ± 1.6 (*p* = 0.005). During this phase, the propranolol group maintained the decrease in ESS (open-label period 4.6 ± 2.05 to 4.18 ± 0.99, *p* = 0.51) and completed a mean drop of 2.76 ± 1.52 in the 16 weeks of treatment (*p* = 0.001). In addition, the entire cohort showed significant improvement in epistaxis control, frequency and severity, and in daily duration of epistaxis (Table 6).

There was no significant improvement in hemoglobin levels (11.42 ± 2.06 to 11.86 ± 1.79, *p* = 0.317) and QOL (41.38 ± 7.38 to 42.92 ± 8.12, *p* = 0.17).

Compliance for the entire cohort during the open-label phase was a mean 78.74 ± 25.2%. Six patients (40%) reported a burning sensation. One participant reported nasal dryness. There was no sign of mucosal damage in any of the participants on rhinology examination, and all chose to continue treatment. No significant change in BP or HR, nor any other side effects, were observed between the beginning and the end of the open label period (systolic BP 110.4 ± 5.9 to 114.6 ± 13.8 mmHg, *p* = 0.27; diastolic BP 63.5 ± 8.4 mmHg to 64.4 ± 9.1 mmHg, *p* = 0.54; HR 72.6 ± 11.8 to 71.6 ± 13 per minute, *p* = 0.55, all respectively).

## 4. Discussion

In a randomized double-blind placebo-controlled study, we demonstrated a significant improvement in epistaxis control in persons with HHT treated with topical propranolol gel for moderate-severe epistaxis. A considerable proportion of persons with HHT have mutations in the endoglin (ENG) and ACVLR1 genes, which encode components of the transforming growth factor (TGF)-β receptor [1]. Recently, a Knudsonian two-hit mechanism was suggested for telangiectases formation [17]. Nonetheless, increased blood and tissue levels of VEGF have been found in persons with HHT [18], suggesting a role for this protein in the abnormal angiogenesis process [6]. The non-selective beta blocker propranolol is used routinely to treat infantile hemangiomas [10,11,19]. The possible mechanisms of action for propranolol include both vasoconstrictive and antiangiogenic effects. The latter are related to VEGF expression, which is controlled by adrenergic stimulation. Propranolol’s antiadrenergic properties directly reduce VEGF-stimulated angiogenesis [11]. In addition, propranolol causes apoptosis of endothelial cells and reduces VEGF tissue expression [11]. Infantile hemangioma resembles HHT in several aspects, including dysregulated angiogenesis and high levels of tissue VEGF [20,21]. These similarities suggest a role for propranolol in the treatment of HHT. Furthermore, propranolol has been demonstrated to decrease in vitro migration and angiogenesis of HHT endothelial cells [22].

Oral and topical propranolol for epistaxis in HHT has been reported in several uncontrolled studies. In a retrospective study, followed by a prospective case series, 21 persons with HHT were treated with oral propranolol, with daily doses of 80–160 mg. In the retrospective arm, a significant decrease in ESS was observed, with improvement in both epistaxis duration and frequency. In the prospective arm, 11 patients were treated with propranolol for three months, which led to a significant decrease in epistaxis duration and in the number of episodes per month. Side effects caused one patient to discontinue propranolol due to hypotension. Other side effects included asthenia, nightmares, and erectile dysfunction [23]. Our group published a case series of six HHT patients treated with nasal propranolol gel. All the patients reported improvement, with a mean decrease of ESS from 6.43 ± 2.11 to 3.47 ± 1.75 during 12 weeks of treatment (*p* = 0.007). Mean hemoglobin levels increased from 8.42 ± 3.06 g/dL to 10.98 ± 1.78 g/dL (*p* = 0.01), and the number of blood transfusions over a 24-week period decreased (*p* = 0.01). One patient reported a burning sensation during the first week of treatment, which resolved during the second week. No other local or systemic effects were observed [12]. A recent observational report demonstrated the efficacy of combined treatment of sclerotherapy and propranolol on epistaxis control in persons with HHT [24]. Anecdotal use of topical timolol has also been reported [25,26].

The targeting of VEGF for treatment of epistaxis in HHT has been attempted with the antibody bevacizumab [27,28,29]. Systemic administration of bevacizumab was found to control nosebleeds and other HHT symptoms [27,28,30,31]. However, systemic bevacizumab caused undesired side effects such as fever, headache, rash, and chills, and notably, epistaxis occurred [32]. Local nasal mucosal injections or spray of bevacizumab have been shown to have a beneficial effect in several case studies and uncontrolled studies, but failed to show an advantage in randomized trials [29]. Recent systematic reviews summarizing the published studies suggest that intranasal bevacizumab treatment does not have a significant effect on epistaxis in persons with HHT [7,8]. Other therapeutic options based on the role of PI3K and mTOR role in telangiectases formation have been recently suggested [33,34]. Severe epistaxis may cause gushing bleeds several times a day, resulting in iron deficiency anemia and transfusion dependency [1]. Response is sometimes inadequate to standard measures of nasal packing, laser coagulation, and arterial embolization. Surgical procedures including septal dermoplasty and Young’s nasal closure may be effective; however, these are partly irreversible and can result in significant morbidity [27].

This randomized placebo-controlled study corroborates previous non-randomized uncontrolled findings of a significant improvement in the ESS score in persons with HHT treated with propranolol. Propranolol was superior to the placebo for improving epistaxis, measured by a significant drop in the ESS. The improvement in epistaxis control with propranolol gel treatment was also demonstrated by the significant increase in HB levels, and decreases in PC transfusions and IV iron requirements in the treatment group but not in the placebo group. These results are strengthened by the significant improvement in epistaxis control in the participants in the placebo group when they were switched to propranolol treatment in the open-label phase. It should be noted that the propranolol group maintained the decrease in ESS during the open label phase, and overall, demonstrated a significant improvement during the 16 weeks of treatment.

Although a one-month run-in period was an option in the protocol, since patients recruited had moderate to severe epistaxis, they were reluctant to have a run-in period. As there was no run-in period, a diary was not available prior to medication administration to measure changes in epistaxis severity, duration, and frequency in the double-blind phase; however, these outcome measures improved in both groups in the open-label phase.

Nasal endoscopy findings in our study groups were defined during follow up visits as no change, improved, or worsened. The rationale for these definitions was based on our experience that improvement was mostly associated with flattening and paling of the lesions and much less often with a change in grading. Improvement in nasal endoscopy findings was observed in 70% of the propranolol and 40% of the placebo group. The improvement in the placebo arm may have been associated with the rigorous regimen of daily nasal lubrication. Daily intranasal application of propranolol gel was associated with improvement in an additional 30% of patients as compared with the placebo group. Although these results were not statistically significant, a larger cohort may show endoscopic improvement in patients treated with topical propranolol. QOL improved in both groups. The improvement in the placebo group could be related to several factors. First, a major component of the SF-12 questionnaire relates to the emotional state of the patient and its influence on their general health. The close follow up and care, the hope for improvement when joining the open-label phase, as well as the moistening effect of the gel, might have contributed to improvement in the patients’ well-being and in QOL. A questionnaire that is more specific to the effects of epistaxis might have shown a statistically significant difference between the groups.

We did not observe any systemic side effects with propranolol treatment. A major side effect was a burning sensation, which affected most of the participants, and led to the withdrawal of three participants from the open-label study. However, the majority of participants reported disappearance or reduction in this sensation with continued treatment. This side effect was less pronounced in the open-label phase. While 40% of the patients reported a burning sensation, all chose to continue treatment, and reported improvement in this side effect. Furthermore, no visible damage was observed in any of the participants in otolaryngologic examination. We believe that propranolol is a mucosal irritant. Since this treatment might be given for longer periods of time, this side effect, as well as other systemic side effects and dose adjustment, should be addressed in future studies.

One patient in the propranolol group and one who withdrew the study from the placebo group developed otitis media. Whether or not methylcellulose gel is the cause is unknown.

Our study is limited due to the relatively small number of patients. Thus, we were unable to reach conclusions regarding some secondary outcomes, as the need for iron infusion. Moreover, important parameters that may affect bleeding such as clinical variants of HHT, type of mutations, as was previously shown by Lesca et al. [35], family relations of the participants, age, and gender were not evaluated. Notably, we had female predominance and, although recent publication did not find a difference in epistaxis frequency between genders [36], a match for gender should be considered in a larger study.

Our novel findings justify and urge a further study with a larger number of patients that could refer to these matters. Further studies should also include a run-in period with an epistaxis diary. The lack of this period in our study precluded assessing the improvement based on these diaries in the placebo-controlled phase.

In summary, our novel findings suggest that topical propranolol gel is safe and effective for epistaxis treatment in patients with HHT-related epistaxis. Larger studies are required to assess the efficacy and safety of systemic and topical propranolol treatment for epistaxis and other HHT related phenomena.

## Figures and Tables

**Figure 1 jcm-09-03130-f001:**
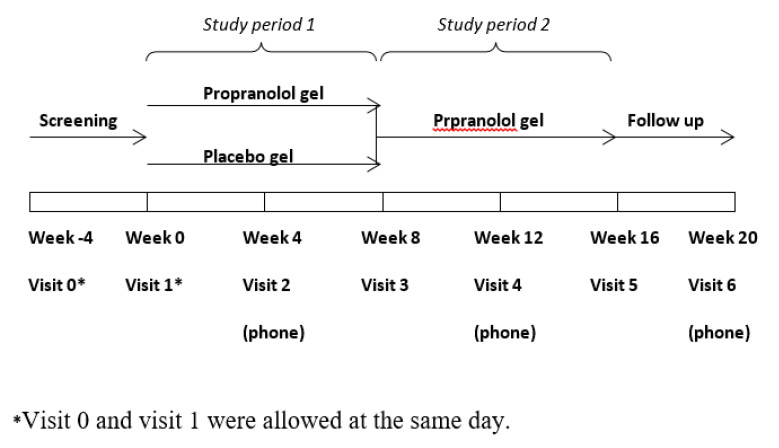
Study design.

**Figure 2 jcm-09-03130-f002:**
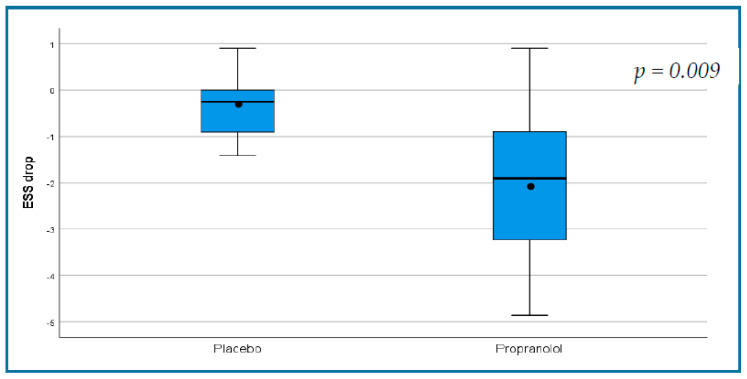
Demonstrating a statistically significant decrease in epistaxis severity score in the propranolol group in the randomized, double-blind phase of the trial.

**Table 1 jcm-09-03130-t001:** Inclusion and exclusion criteria.

Inclusion criteria: Adults, 18 years and older Confirmed clinical [13] or genetic diagnosis of HHT Epistaxis severity score (ESS, range of scores: 0–10, 10 indicates greatest severity) [14]) ≥4 Refractory to standard measures that control bleeding
Exclusion criteria: Congestive heart failure, baseline bradycardia (heart rate (HR) < 50/min), heart block, asthma or known sensitivity to propranolol). Treatment with beta-blockers for other reasons Change in epistaxis care in the month prior to enrolment Use of any antiangiogenic medication in the month prior to enrolment Use of antiplatelet or anticoagulant medications

**Table 2 jcm-09-03130-t002:** Baseline characteristics.

	Placebo (*n* = 10)	Propranolol (*n* = 10)	*p*
Age—years (mean ± SD)	51 ± 9	57 ± 11	0.262
Gender (F,M)	9:1	6:4	0.152
Gene mutation (number of participants)	ACVRL1-8 Endoglin-1 ND-1	ACVRL1-5 Endoglin-3 ND-2	0.223
ESS	5.68 ± 1.8	6.50 ± 1.84	0.323
QOL	34.75 ± 10.9	35.1 ± 6.7	0.932
Hemoglobin (g/dL)	10.7 ± 2.52	10.57 ± 2.6	0.918
IV iron */IV PC *	16/3	9/7	0.739/0.481
Ferritin	65.4 ± 86.19	89.65 ± 212.04	0.715
Rhinology grading (number of patients)			0.601
Grade I	4	3	
Grade II	5	5	
Grade III	1	2	
Systolic BP mmHg (mean ± SD)	110.2 ± 9.6	119.3 ± 12.3	0.09
Diastolic BP mmHg (mean ± SD)	63.3 ± 9.2	69.0 ± 9.2	0.20
HR per minute (mean ± SD)	76.0 ± 12.9	73.9 ± 111.7	0.71

* Total number of units of IV iron (Ferinject 500 mg)/IV packed cells (PC) in the 2 months prior to screening. ESS, epistaxis severity score; ACVRL1, activin receptor-like 1; ND, not done; QOL, quality of life; IV, intravenous; PC, packed cells; BP, blood pressure; HR, heart rate.

**Table 3 jcm-09-03130-t003:** Outcome measures during the double-blind phase of the study. (Placebo group and propranolol group are analyzed separately).

Outcome Measure	Placebo (*n* = 10)			Propranolol (*n* = 10)		
	Baseline	End of DB Phase	*p*	Baseline	End of DB Phase	*p*
Primary outcome						
ESS	5.68 ± 1.8	5.33 ± 2.1	0.133	6.50 ± 1.84	4.47 ± 1.75	0.004
Secondary outcomes						
Hb g/dL	10.7 ± 2.5	10.7 ± 2.3	0.91	10.5 ± 2.6	11.4 ± 2.02	<0.001
Total PC units required *	3	5	0.346	7	3	<0.001
Total IV iron **	16	14	0.233	9	4	0.15
doses required *						
QOL	34.75 ± 10.9	40.6 ± 9.11	0.03	35.1 ± 6.7	41 ± 7.39	0.048

ESS, epistaxis severity score; DB, double-blind; Hb, hemoglobin; PC, packed cells; IV, intravenous; QOL, quality of life. * In the 2 prior months and ** dose, Ferinject 500 mg.

**Table 4 jcm-09-03130-t004:** Comparison of the change in outcome measures during the double-blind period among the placebo and propranolol groups.

Outcome Measure	Placebo	Propranolol	*p*
Primary outcome			
Change in ESS	−0.35 ± 0.68	−2.03 ± 1.7	0.009
Secondary outcomes			
Change in Hb level (g/dL)	0.68 ± 0.01	1.96 ± 0.85	0.216
Change in number of PC required	−0.20 ± 0.63	−0.40 ± 0.69	0.029
Change in number of IV iron doses required *	−0.20 ± 0.63	−0.50 ± 0.97	0.304
QOL	6.11 ± 5.85	6.06 ± 5.90	0.986

ESS, epistaxis severity score; Hb, hemoglobin; PC, packed cells; IV, intravenous; QOL, quality of life. * Dose, Ferinject 500 mg.

**Table 5 jcm-09-03130-t005:** Side effects of the double-blind period.

	Placebo (10)	Propranolol (10)	*p*
Any burning sensation	2	9	0.005
Sustained burning sensation	0	4	0.086
Rhinorrhea	0	3	0.06
Nasal dryness	1	0	1
Otitis media	0	1	1

**Table 6 jcm-09-03130-t006:** Diary outcome measures in the open label phase, diary comparison.

Outcome Measure (mean ± SD) *n* = 15	Double Blind Period	Open-Label Period	*p*
Epistaxis frequency, bleeds/day	1.71 ± 1.34	1.24 ± 1.24	<0.001
Epistaxis severity	1.42 ± 0.64	1.03 ± 0.55	<0.001
Epistaxis duration, minutes/day	10.68 ± 9.01	6.13 ± 4.67	<0.001

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
