# Peer review of "Topical Propranolol Improves Epistaxis Control in Hereditary Hemorrhagic Telangiectasia (HHT): A Randomized Double-Blind Placebo-Controlled Trial"

_jcm, 2020, doi:10.3390/jcm9103130_

Round 1

Reviewer 1 Report

The authors report on the use of propanolol in patients affected with HHT. I suggest, both in title and text to write "improvement in epistaxis control " (or something similar) and not just "improvement of epistaxis" at least a short comment about the presence/absence of intestinal bleeding should be added when talking about recruited patients (it is more common in older patients, and may largely influence Hb levels) it should be clearly stated that (if) these patients are additional to the small group already studied by the authors. although a calculation of sample size was perfomed, due to the large clinical variation, I feel that a larger sample should be studied the genetic status is not available in all cases; as the genetic status clearly influences several parameters of epistaxis (lesca et al 2007) (you may be willing to quote this paper), this data must be obtained. state that patients were unrelated, or detail if some of them belong to the same family major concern - small sample size - I feel that matching cases on propanolol or placebo for sex, age, gene involved, severity score would provide much better result Discussion: the sentence: "The association of these mutations to telangiectasis formation is unknown. " like it is is not clear, please change it please quote the recent paper by Gallione et al which identifies the double hit mechanism in the pathogenesis of the disease. the sentence: " Although ESS improvement was not statistically significant in the open-label phase, overall, a significant improvement was observed in the 16 weeks 261 of treatment. " is not clear: the autrhors, just before, state about the significant improve for ESS for cases previously on placebo. please make it more clear about rinoscopy, the picture included shows very nice results, however I understand from the text that this is not the commonest result obtained; probably it is worthwile to provide also a picture with the milder and more common outcomes.

Author Response

We would like to thank the reviewer for the important comments which helped us improving our manuscript. Please see our answers to each comment.

The authors report on the use of propanolol in patients affected with HHT.

*I suggest, both in title and text to write "improvement in epistaxis control " (or something similar) and not just "improvement of epistaxis"- Thank you for this correction- The title was changed (line 1), as well as in the text (line 305, 321, 379, 382)

*at least a short comment about the presence/absence of intestinal bleeding should be added when talking about recruited patients (it is more common in older patients, and may largely influence Hb levels) it should be clearly stated that (if) these patients are additional to the small group already studied by the authors. This is an important point that was not mentioned in the methods- patients with significant GI bleed were not included and assessed for systemic treatment. We added this sentence – line 77-78    "Patients with concomitant significant gastrointestinal bleeding were referred for systemic treatment and were not included in the study".

*although a calculation of sample size was perfomed, due to the large clinical variation, I feel that a larger sample should be studied – Thank you – We agree that the number of patients in the study is relatively small, which did not enable us to analyze the results according to clinical variants of HHT. However please note that the number of patients needed for the study was calculated ahead of study to answer the primary outcome of differences in the ESS score in HHT patients (all patients diagnosed with HHT according to the Curacao criteria), as described in the experimental section (line 160)

We do believe that our findings justify and urge a further study with a larger number of patients to further study the beneficial effect of topical propranolol in HHT patients according to clinical variants.

This is better discussed now, please see the discussion section (from line 428)

*the genetic status is not available in all cases; as the genetic status clearly influences several parameters of epistaxis (lesca et al 2007) (you may be willing to quote this paper), this data must be obtained. state that patients were unrelated, or detail if some of them belong to the same family. major concern - small sample size - I feel that matching cases on propanolol or placebo for sex, age, gene involved, severity score would provide much better result –

Thank you, indeed all these questions are highly important. However, since they were not part of our endpoints, the number of patients could not target any answers. We do know the genetic diagnosis of 17/20 patients and know that two of the participants were family related. But unfortunately, as stated, the numbers to conclude are too low, This is now better discussed, please see the discussion section (including the ref) (from line 428). 

*Discussion: the sentence: "The association of these mutations to telangiectasis formation is unknown. " like it is is not clear, please change it please quote the recent paper by Gallione et al which identifies the double hit mechanism in the pathogenesis of the disease- thank you for this important reference- it was added- line 326. "Recently a Knudsonian two-hit mechanism was suggested for telangiectases formation"

*the sentence: "Although ESS improvement was not statistically significant in the (propranolol group in the)  open-label phase, overall, a significant improvement was observed in the 16 weeks  of treatment. " is not clear: the autrhors, just before, state about the significant improve for ESS for cases previously on placebo. please make it more clear –

Thank you – the sentence was not clear – it regards to the propranolol group in the open label period which sustained the improvement but did not continue to drop the ESS. We did rephrase the sentence:

"It should be noted that the propranolol group maintained the decrease in ESS during the open label phase, and overall, demonstrated a significant improvement in the 16 weeks of treatment".. from line 383. *about rinoscopy, the picture included shows very nice results, however I understand from the text that this is not the commonest result obtained; probably it is worthwile to provide also a picture with the milder and more common outcomes.- Thank you - 70% of all and 71.4% of the moderate-severe epistaxis improved on propranolol but it was not significant as there was some improvement (40% and 16% respectively) in the placebo group. We believe that with larger numbers it might be significant. However – we agree that since the results are not significant and since other reviewers suggested modifying the rhinoscopy presentation, we should omit the picture.

Reviewer 2 Report

This is a small randomized clinical trial of topical propranolol in a gel formulation compared to placebo for moderate to severe HHT related epistaxis.  The study follows up a previous open label study that observed a meaningful decrease in epistaxis severity with open label therapy.

The clinical problem is important, and existing treatments inadequate.  This study is of great interest.

To summarize the findings:

Power calculations estimated a need for 9 patients and 9 controls to complete the study, enrolment was intended to recruit 11 for each arm.  Out of 23 patients recruited, 10 received active treatment, 13 received placebo, 3 did not complete the protocol (all from placebo arm).

The primary outcome was change in ESS from baseline to end of 8-week treatment (double-blind phase). Secondary outcomes included change in hemoglobin levels, requirement for IV iron or blood transfusion, visible changes to nasal telangiectasias by endoscopy, and quality of life.

Patients were well matched for baseline characteristics. Treatment with propranolol resulted in a significant improvement in the primary endpoint – reduction in epistaxis severity score – among treated patients, with no significant change in the placebo group.

Favorable changes were also observed for certain secondary outcomes including improvement in hemoglobin (treatment group only), number of patients requiring transfusion (treatment group only) and number of units of blood transfused (treatment group only).  The number of patients requiring IV iron also improved in the treatment group, but not the number of iron doses required.  Curiously, quality of life improved in both groups to a similar degree.

No statistically significant differences were observed by nasal endoscopy.

Propranolol gel was associated with a burning sensation in the nasal mucosa and also rhinorrhea for treated patients.  These symptoms contributed to a 30% drop out rate after 8 weeks of therapy.  Introduction of a nasal gel may have contributed to acute otitis media seen in both groups.

Open label therapy was provided to all willing participants for a further 8 weeks.  The improvement in ESS was maintained for the treatment group (no statistical comparison provided for the final ESS relative to 8 week ESS in this subset).  The former placebo patients saw an improvement in ESS, but the statistical comparison to baseline or to 8 week ESS is not provided in the manuscript.  30% of the patients previously on active treatment chose to not continue therapy.

The daily epistaxis diary provided data to explore the impact of propranolol on epistaxis frequency, severity and duration, but only in comparison between the end of the 8-week randomized phase and the 16-week open label phase because of a lack of baseline data and no run-in period for the trial.

Critique:

This report is an important advance, and it appears that topical propranolol offers a short-term meaningful benefit for HHT related epistaxis, with potential risks.  The durability of benefit and the safety of therapy are not well defined.  More clarity is requested regarding trial logistics and outcome measures.  There are also questions regarding the statistical analysis.

Trial logistics:

  1. How many patients were screened for trial eligibility? This is important to provide context for the frequency that HHT centers might encounter these patients.
  2. The recruitment, randomization, drop-out in the randomized phase, and drop-out before the open-label phase are described in the text, but the addition of a CONSORT diagram for the trial would allow a more rapid appreciation for the trial logistics.
  3. It is unfortunate that the data from the nosebleed diary was not a very useful outcome measure because of the lack of run-in data. The analysis (Table 4, lines 203-204) is confusing.
    1. What period of time was used to calculate the daily epistaxis frequency, severity or duration at the “Start of open-label phase” and “End of open-label phase”?
    2. While one might presume that this flaw should have been apparent to the investigators when designing the trial, since it wasn’t appreciated, there should be more discussion about this shortcoming to better inform designers of future trials to include a run-in period for better baseline data.
    3. How was the analysis actually done? This appears to be a description of results only for the open-label, non-randomized phase.
      1. The “baseline” data here is the “Start of open-label phase” which apparently includes patients treated with propranolol mixed together with placebo patients. Is this a true assumption?
      2. Did the “Start of open label phase” data include all patients in the randomized phase, or was it restricted to those who were participating in the open-label phase?
  • Could the data be compared by the prior randomized treatment group, only for those participating in the open-label phase? This is no longer a blinded treatment status, which is clearly problematic, but one could report the impact of going from placebo to active treatment (expect to see benefit) and separately the impact of staying on therapy but becoming aware of treatment group assignment (durability of response, some measure of placebo effect).
  1. The nosebleed diary allows assessment of the individual components of epistaxis severity: frequency of bleeds, duration of independent bleeds, and severity of bleeding episodes. The data are reported rather with the cumulative duration of bleeding that is impacted also by the frequency of bleeding.  Although reducing this cumulative daily duration may be the ultimate goal for any therapy, it would be useful to report and analyze the impact of propranolol on individual bleeding episode duration (divide total duration by frequency).  A quick assessment of the data from the table suggests that propranolol may have an impact on both frequency and individual bleed duration.
    1. The validity of these statistical comparisons should be reviewed by a knowledgeable editor or reviewer.

Outcome measures:

  1. The primary outcome as described is ambiguous (lines 103-104). The power calculation, the figure (Figure 1), and the statistical comparison are all based on ESS compared from baseline to end of 8 weeks of blinded therapy. This must be made clearer. The ESS from end of blinded phase to end of open label phase is a secondary outcome.
  2. The nasal endoscopy data has issues with presentation and interpretation.
    1. The difference between groups was not statistically significant. The paper should not infer that there was a visible difference (see lines 27&28, lines 161-163, Figure 2) without statistical significance.
  3. Very little data is actually presented comparing the groups at the end of the 8-week randomized phase to the end of the open label phase for these secondary outcomes. This could be a separate table, comparable to Table 3, or included as a third column with the data presented in table 3.

Statistical analysis.

  1. The statistical analyses should be reviewed by a knowledgeable editor or statistical reviewer. My concerns are primarily:
    1. Why was the primary and secondary outcome analysis performed comparing post-treatment to baseline data within group? I would expect to see a direct comparison of treatment group to placebo group for each outcome.
    2. The Rhinology scoring was assessed with the McNemar test, which explains the presentation of a single P value for the data set. A McNemar test compares a dichotomous variable between 2 groups (2 x 2 table format), yet the dependent variable has 3 categories between the two groups.  How did the reviewers divide the rhinology assessment into 2 categories for comparison by this test?
    3. Table 3, Number of patients requiring IV Fe: how can a change from 4 to 3 patients in the placebo group have a P value of 0.8 and a change in the treatment group from 5 to 3 have a P value of 0.03?
    4. How should the nosebleed diary data be statistically analyzed? (see point 3.d. above)

Other data presentation issues:

  1. Table 1. No issues
  2. Table 2.
    1. The 9th row of the table lists “0.739/0.481” in each column, without explanation. It is thought this is a typographical error, and should be omitted.
  3. Figure 1.
    1. The figure is not easily interpreted independent of the text of the article. This figure is the key figure for the paper – a statistically significant, clinically meaningful decrease in epistaxis severity in the randomized, double-blind portion of the trial. This is the primary outcome, and this figure should be independently interpretable even if the rest of the paper is not reviewed in detail. 
      1. The figure legend should be more descriptive.
      2. It could be helpful to indicate what reduction in ESS is clinically meaningful.
  • There is no description of statistical significance in the figure.
  1. Table 3.
    1. There is unusual variation in font size
    2. There is a heading line that extends only partially across the figure (distracting).
    3. The data for nasal endoscopy is not included, but was mentioned also as a secondary outcome. It should be included here, not just in Figure 2.
    4. This table includes both primary and secondary outcome measures. I would like to see the outcomes labeled as such, perhaps with a line subdividing the figure into primary outcome (ESS) and all secondary outcomes grouped together below.
  2. Figure 2.
    1. This is purely descriptive, anecdotal data, and do not represent the typical response. A more truthful presentation of the data would show a comparison sample most closely representing the median response.
    2. Given that this is a small trial, a figure showing before and after imaging of all participants would be interesting as a supplemental figure.
  3. Table 4. (see point 3 above)
  4. Adverse effects of therapy. There should be a table dedicated to adverse effects, even if purely descriptive.  These topics should also be specifically addressed in the discussion section.
    1. It appears that propranolol in this preparation is likely a nasal irritant (burning sensation, rhinorrhea), and long-term use may be necessary before seeing endoscopic changes.
    2. It is possible that the methylcellulose gel is a trigger for acute otitis media.
    3. There should be a disclosure of beta-blocker outcomes. What was the baseline heart rate and blood pressure, and what was the heart rate and blood pressure at the 8-week and 16-week clinical assessment time points?

Author Response

We thank the reviewer for the comprehensive review. These are our repnses to each comment:

Open label therapy was provided to all willing participants for a further 8 weeks.  The improvement in ESS was maintained for the treatment group (no statistical comparison provided for the final ESS relative to 8 week ESS in this subset). Thank you. this information was added to the text. Line267: During this phase, the propranolol group maintained the decrease in ESS (open label period- 4.6±2.05 to 4.18±0.99, p=0.51) and completed a mean drop of 2.76±1.52 in the 16 weeks of treatment (p=0.001).

The former placebo patients saw an improvement in ESS, but the statistical comparison to baseline or to 8 week ESS is not provided in the manuscript. Thank you. this information was added to the text. Line 275 During the open-label phase, the former placebo group, now receiving propranolol gel, showed significant improvement in the ESS score, with a mean drop of 1.99±1.41 from 4.86±1.85 to 2.87±1.6 (p=0.005).

 30% of the patients previously on active treatment chose to not continue therapy.

The daily epistaxis diary provided data to explore the impact of propranolol on epistaxis frequency, severity and duration, but only in comparison between the end of the 8-week randomized phase and the 16-week open label phase because of a lack of baseline data and no run-in period for the trial.

Critique:

This report is an important advance, and it appears that topical propranolol offers a short-term meaningful benefit for HHT related epistaxis, with potential risks.  The durability of benefit and the safety of therapy are not well defined.  More clarity is requested regarding trial logistics and outcome   measures.  There are also questions regarding the statistical analysis.

Trial logistics:

  1. How many patients were screened for trial eligibility? This is important to provide context for the frequency that HHT centers might encounter these patients. Thank you – a CONSORT was added in a separate file. It can be inserted to the draft or presented in a supplementary file
  2. The recruitment, randomization, drop-out in the randomized phase, and drop-out before the open-label phase are described in the text, but the addition of a CONSORT diagram for the trial would allow a more rapid appreciation for the trial logistics. We added a figure demonstrating the design. Line 
  3. It is unfortunate that the data from the nosebleed diary was not a very useful outcome measure because of the lack of run-in data. The analysis (Table 4, lines 203-204) is confusing. Although we had an option of 1 month run-in period- patients (experiencing severe epistaxis) were reluctant to wait a month. We are aware to the fact that the diary was not useful for the double-blind period. It does show improvement in the open label period in all patients.
    1. What period of time was used to calculate the daily epistaxis frequency, severity or duration at the “Start of open-label phase” and “End of open-label phase”?
    2. While one might presume that this flaw should have been apparent to the investigators when designing the trial, since it wasn’t appreciated, there should be more discussion about this shortcoming to better inform designers of future trials to include a run-in period for better baseline data. The table was modified to make it clearer . We added in the discussion – line 389:"Although a one month run-in period was an option in the protocol -since patients recruited had moderate to severe epistaxis, they were reluctant to have a run-in period". This is mentioned as a limitation and should be addressed in a future study
    3. How was the analysis actually done? This appears to be a description of results only for the open-label, non-randomized phase.
      1. The “baseline” data here is the “Start of open-label phase” which apparently includes patients treated with propranolol mixed together with placebo patients. Is this a true assumption? Yes. The numbers are a mean and SD of each period. We changed the table accordingly. We do have the results of each group but since both got the same treatment, we presented the whole group. If the reviewer thinks it is better to present the data separately it can be added to the manuscript.
      2. Did the “Start of open label phase” data include all patients in the randomized phase, or was it restricted to those who were participating in the open-label phase? Only the ones that continue the open label period
  • Could the data be compared by the prior randomized treatment group, only for those participating in the open-label phase? This is no longer a blinded treatment status, which is clearly problematic, but one could report the impact of going from placebo to active treatment (expect to see benefit) and separately the impact of staying on therapy but becoming aware of treatment group assignment (durability of response, some measure of placebo effect). The statistics of the open-label phase, for both the former placebo group, now receiving propranolol gel, and the propranolol group was only for the patients continuing treatment. Patients who resigned were not included. LINE 300 " During the open-label phase, the former placebo group, now receiving propranolol gel, showed significant improvement in the ESS score, with a mean drop of 1.99±1.41 from 4.86±1.85 to 2.87±1.6 (p=0.005). During this phase, the propranolol group maintained the decrease in ESS (open-label period- 4.6±2.05 to 4.18±0.99, p=0.51) and completed a mean drop of 2.76±1.52 in the 16 weeks of treatment (p=0.001)"
  1. The nosebleed diary allows assessment of the individual components of epistaxis severity: frequency of bleeds, duration of independent bleeds, and severity of bleeding episodes. The data are reported rather with the cumulative duration of bleeding that is impacted also by the frequency of bleeding.  Although reducing this cumulative daily duration may be the ultimate goal for any therapy, it would be useful to report and analyze the impact of propranolol on individual bleeding episode duration (divide total duration by frequency).  A quick assessment of the data from the table suggests that propranolol may have an impact on both frequency and individual bleed duration.
    1. The validity of these statistical comparisons should be reviewed by a knowledgeable editor or . We looked at this interesting point and calculated the duration of each episode. There was a large variability but the difference in the length of each episode was not significant p=0.24

Outcome measures:

The primary outcome as described is ambiguous (lines 103-104). The power calculation, the figure (Figure 1), and the statistical comparison are all based on ESS compared from baseline to end of 8 weeks of blinded therapy. This must be made clearer. The ESS from end of blinded phase to end of open label phase is a secondary outcome. Thank you very much – it is a very important point. We made the changes- LINE 147: " The primary outcome was the difference in ESS drop between both groups: ESS was first calculated at randomization and was related to the 8 weeks prior to randomization, and then at the end of the double-blind period. The change in ESS from randomization to the end of the first 8-week period was compared between the study and the control groups. And for the open label period: " In the open label period the secondary outcome was ESS change from the beginning and the end of the open label phase".

  1. The nasal endoscopy data has issues with presentation and interpretation.
    1. The difference between groups was not statistically significant. The paper should not infer that there was a visible difference (see lines 27&28, lines 161-163, Figure 2) without statistical significance. The figure and these sentences were omitted
  2. Very little data is actually presented comparing the groups at the end of the 8-week randomized phase to the end of the open label phase for these secondary outcomes. This could be a separate table, comparable to Table 3, or included as a third column with the data presented in table 3. We mainly looked at the secondary outcomes for the double-blind period. Some of the sec outcomes were added to the text. Due to the short period of time some of the data could not be added as they need reviewing the charts.

Statistical analysis.

  1. The statistical analyses should be reviewed by a knowledgeable editor or statistical reviewer. My concerns are primarily:
    1. Why was the primary and secondary outcome analysis performed comparing post-treatment to baseline data within group? I would expect to see a direct comparison of treatment group to placebo group for each outcome. The primary outcome was presented both ways (table 3 and figure 2 and now also table 4 which was added to compare the sec outcomes between the groups.   

The Rhinology scoring was assessed with the McNemar test, which explains the presentation of a single P value for the data set. A McNemar test compares a dichotomous variable between 2 groups (2 x 2 table format), yet the dependent variable has 3 categories between the two groups.  How did the reviewers divide the rhinology assessment into 2 categories for comparison by this test? Thank you very much for your comment – the statistical analysis was not clearly described in the method. After grading the change to improved / no change / worsen – the results were classified as improved vs did not improved- and McNamer test was the test performed. We changed the definition in the methods section to:LINE 113 During follow up, endoscopies were defined as –  improved versus no change or worsened.

    1. Table 3, Number of patients requiring IV Fe: how can a change from 4 to 3 patients in the placebo group have a P value of 0.8 and a change in the treatment group from 5 to 3 have a P value of 0.03?
    2. How should the nosebleed diary data be statistically analyzed? (see point 3.d. above) see our comment above.

Other data presentation issues:

  1. Table 1. No issues
  2. Table 2.
    1. The 9th row of the table lists “0.739/0.481” in each column, without explanation. It is thought this is a typographical error, and should be omitted. Thank you. We apologize – it is a typo. The right line was inserted-

IV Fe /IV PC*

16/3

9/7

0.739 / 0.481

    1.  
  1. Figure 1.
    1. The figure is not easily interpreted independent of the text of the article. This figure is the key figure for the paper – a statistically significant, clinically meaningful decrease in epistaxis severity in the randomized, double-blind portion of the trial. This is the primary outcome, and this figure should be independently interpretable even if the rest of the paper is not reviewed in detail. 
      1. The figure legend should be more descriptive.Thank you – it was changed to – LINE 119 Figure 1- demonstrating a statistically significant decrease in epistaxis severity score (ESS) in the randomized, double-blind portion of the trial 

It could be helpful to indicate what reduction in ESS is clinically meaningful. Thank you – a reference was added- "ESS change of 0.71 was suggested to be the MID (minimal important difference)" (Yin- ref 36)

      1.  
  • There is no description of statistical significance in the figure.- thank you – it was added to the figure.
  1. Table 3.
    1. There is unusual variation in font size- thank you – it was changed
    2. There is a heading line that extends only partially across the figure (distracting).
    3. The data for nasal endoscopy is not included, but was mentioned also as a secondary outcome. It should be included here, not just in Figure 2.- the results of the nasal endoscopy are different as we looked at number improved and not at two values at start and end. We can prepare a different table for this outcome measure, however since results were not statistically significant, we did not build another table. It can be added if the reviewer thinks it is valuable.
    4. This table includes both primary and secondary outcome measures. I would like to see the outcomes labeled as such, perhaps with a line subdividing the figure into primary outcome (ESS) and all secondary outcomes grouped together below.Thank you – table was modified
  2. Figure 2.
    1. This is purely descriptive, anecdotal data, and do not represent the typical response. A more truthful presentation of the data would show a comparison sample most closely representing the median response-Thank you- figure was deleted.
    2. Given that this is a small trial, a figure showing before and after imaging of all participants would be interesting as a supplemental figure. thank you for the idea- we can prepare a file with all figures- if possible, it can be added as a supplement
  3. Table 4. (see point 3 above)
  4. Adverse effects of therapy. There should be a table dedicated to adverse effects, even if purely descriptive.  These topics should also be specifically addressed in the discussion section. A table was added. Line 291
    1. It appears that propranolol in this preparation is likely a nasal irritant (burning sensation, rhinorrhea), and long-term use may be necessary before seeing endoscopic changes. We totally agree that it is an important issue. It was added to the discussion- - LINE 422 We believe that propranolol is a mucosal irritant and this side effect should be addressed in future studies.  
    2.  

It is possible that the methylcellulose gel is a trigger for acute otitis media. Thank you – we added this sentence- One patient at each group developed otitis media and whether methylcellulose gel is the cause has to be further assessed.

    1. There should be a disclosure of beta-blocker outcomes. What was the baseline heart rate and blood pressure, and what was the heart rate and blood pressure at the 8-week and 16-week clinical assessment time points? Baseline BP and HR was added to table 1. The values at 8 weeks are presented in LINE 275. The results at 16 weeks were added to the text. LINE 313

Reviewer 3 Report

Journal of Clinical Medicine

Topical Propranolol Improves Epistaxis in Hereditary Hemorrhagic Telangiectasia (HHT): A Randomized Double-blind Placebo-Controlled Trial. Meir Mei-Zahav, Yulia Gendler, Elchanan Bruckheimer, Dario Prais, Eynat Birk, Muhamad Watad, Neta Goldschmidt and Ethan Soudry.

In this manuscript, Mei-Zahav M et al. investigated efficacy and safety of 0.5 cc propranolol gel twice daily to the nasal mucosa of each nostril in a two-phase RCT: a double-blind placebo-controlled phase and an open-label phase. The question being addressed here is very important in the context of HHT, because epistaxis is the most common clinical symptom of HHT patients, there is no curative treatment for nosebleedings and epistaxis is associated with significant impairment quality of life. The paper is well written and contains useful information. In general, the authors deploy a good methodology to achieve the main goals. The objective of the paper could be very useful for clinicians dealing with HHT patients, however the following points should be addressed in order to improve the final message:

GENERAL:

1.- The dilated post capillary venules directly connected with dilated arterioles losing the capillary bed is called telangiectasis. The plural of telangiectasis is telangiectases and the presence of telangiectases is called telangiectasia. In this document, the word telangiectasia is used both for single and plural. You should change that in the document.

ABSTRACT:

2.- I suggest to divide the abstract in different sections (Background, Methods, Results and Conclusions).

INTRODUCTION:

3.- Please add the ORPHA number of HHT (ORPHA774) and the OMIM numbers for HHT1 and HHT2 (HHT 1: OMIM# 187300; HHT 2: OMIM# 600376).

METHODS:

4.- This reviewer misses some more information about the statistical methods. Please indicate in this section the method used to assess normality.

5.- Are patients with antiplatelet or anticoagulant treatment included? It should be written in the exclusion criteria.

6.- I think it would be of great help including a figure with the timeline where readers can rapidly see how was the study conducted (screening, visits, assessment, withdrawal, end of the blinded period and beginning of the open-label, treatment switch,…). As a basic example:

RESULTS:

7.- Please provide appropriate abbreviations in each Table.

8.- In Table 2 there is a typographic mistake (0.739/0.481). Also in table 2, the asterisk in the legend does not correspond to anything.

9.- Did you collect the need for Emergency Department due to nosebleeds during the study?

DISCUSSION:

10.- Some sentences of this section are previously mentioned in the Introduction or Results sections. Please reduce this section eliminating these repeated concepts and discuss other important ones.

11.- The authors should discuss in this section if the high proportion of females could influence the results. Please, review a recent paper regarding gender differences in HHT severity (in which no differences were found in ESS between men and women).

12.- Please briefly discuss other possible therapeutic mechanisms for HHT such as PI3K or mTOR inhibitors. There is some evidence in animal models and human telangiectases (see Iriarte et al. Cells 2019)

13.- Caution should be used here by mentioning the possible long-term side-effects or the need for dose-adjustments for propanolol, especially when considering that patients might need to be treated for many years, if not decades.

14.- Line 210, the sentence “the association of these mutations to telengiectasis formation is unknown” is not really true. The association is clear; what is not understood is the mechanism.

Author Response

We thank the reviewer for the comprehensive review. Please find our answers to each comment:

GENERAL:

1.- The dilated post capillary venules directly connected with dilated arterioles losing the capillary bed is called telangiectasis. The plural of telangiectasis is telangiectases and the presence of telangiectases is called telangiectasia. In this document, the word telangiectasia is used both for single and plural. You should change that in the document.- Thank you – these correction were made.

ABSTRACT:

2.- I suggest to divide the abstract in different sections (Background, Methods, Results and Conclusions). – Thank you – the journal guidelines are-  "The abstract should be a single paragraph and should follow the style of structured abstracts, but without headings" https://www.mdpi.com/journal/jcm/instructions

INTRODUCTION:

3.- Please add the ORPHA number of HHT (ORPHA774) and the OMIM numbers for HHT1 and HHT2 (HHT 1: OMIM# 187300; HHT 2: OMIM# 600376). Thank you. These numbers have been added to the text. Line-

METHODS:

4.- This reviewer misses some more information about the statistical methods. Please indicate in this section the method used to assess normality. Thank you - this important sentence was added to the statistical methods: Line: "Normality assessment was performed using a Shapiro Wilk test. Normal distribution was assumed when p>0.05".

5.- Are patients with antiplatelet or anticoagulant treatment included? It should be written in the exclusion criteria. Thank you for the comment. The protocol required patients to be on stable treatment with no change in treatment for 3 months prior to the study. 2 patients in each group were on transexamic acid as a chronic treatment and were asked to stay on the same regimen

6.- I think it would be of great help including a figure with the timeline where readers can rapidly see how was the study conducted (screening, visits, assessment, withdrawal, end of the blinded period and beginning of the open-label, treatment switch,…). As a basic example: Thank you. A figure was added. Line ….

RESULTS:

7.- Please provide appropriate abbreviations in each Table. Thank you – abbreviations were added to each table

8.- In Table 2 there is a typographic mistake (0.739/0.481). Also in table 2, the asterisk in the legend does not correspond to anything. Thank you . we apologize for the typo mistake- Table was modified due to other comments from reviewers

9.- Did you collect the need for Emergency Department due to nosebleeds during the study?

Thank you. A very important point. Yes. We have these data however we did not analyze as it was not an outcome. We can extract these data (will require few more days- review time is very short)

DISCUSSION:

10.- Some sentences of this section are previously mentioned in the Introduction or Results sections. Please reduce this section eliminating these repeated concepts and discuss other important ones. Thank you – we reviewed the introduction and discussion and deleted few sentences e.g " Infantile hemangioma resembles HHT in several aspects, including dysregulated angiogenesis and high levels of tissue VEGF" in the introduction.

11.- The authors should discuss in this section if the high proportion of females could influence the results. Please, review a recent paper regarding gender differences in HHT severity (in which no differences were found in ESS between men and women). Thank you – a short discussion was added with the relevant reference. Line 433 "We had female predominance and although recent publication did not find difference in epistaxis frequency between genders (Mora-Luian) a match for gender should be considered in a larger study".

12.- Please briefly discuss other possible therapeutic mechanisms for HHT such as PI3K or mTOR inhibitors. There is some evidence in animal models and human telangiectases (see Iriarte et al. Cells 2019) Thank you – we added with the references LINE 367– " Other therapeutic options based on the role of PI3K and mTOR role in telangiectases formation have been recently suggested .

13.- Caution should be used here by mentioning the possible long-term side-effects or the need for dose-adjustments for propanolol, especially when considering that patients might need to be treated for many years, if not decades.Thank you for this important issue: We added. LINE 423:" Since this treatment might be given for longer periods of time, this side effect as well as other systemic side effects and dose adjustment should be addressed in future studies".

14.- Line 210, the sentence “the association of these mutations to telengiectasis formation is unknown” is not really true. The association is clear; what is not understood is the mechanism.Thank you – this sentence was omitted. The 2 hit mechanism theory was inserted. Line 325: "Recently a Knudsonian two-hit mechanism was suggested for telangiectases formation".

Reviewer 4 Report

Minor comments

  1. There is some inconsistency in the use of telangiectasia, telangiectasis, and telangiectases.
  2. Line 70. Do we know anything about the timeline of response with propranolol gel and whether 8 weeks of treatment was optimal to detect a response?
  3. Line 87. How was the determination of change made re endoscopy findings?  Were photos or videos taken at each timepoint and then compared?  Was it based on memory of the last endoscopy?  Based on notes?
  4. Line 132. I do not understand the line below hemoglobin in table 2. The same numbers repeat in each column.
  5. Line 154. Epistaxis has been consistently shown to be the biggest determinant of QOL in HHT patients. Can the authors please discuss why placebo and drug patients showed a similar improvement in QOL despite a significant improvement in epistaxis being seen only in drug patients?
  6. Line 164, Figure 2. Cell “A” is mentioned in the legend but not in the figure.
  7. Please comment on whether the burning sensation may have been a subconscious clue to patients that they were receiving active drug and could have affected perceived response. Was there any correlation between burning and effect on epistaxis?

Major comments:

  1. Line 61. Use of antiangiogenics in the past 1 month is listed as exclusionary. Use of systemic pazopanib or bevacizumab in the past 3 months could potentially influence results due to known prolonged time course of response.  Did any patients receive either of these in the prior 3 months?
  2. Line 125. It is interesting that at study end there were 3 more placebo patients recruited relative to drug (13 vs 10), and that 3 placebo dropped out to give a final count of 10 in each group.  Did the randomization process somehow track this and attempt to maintain equal final distribution of the groups?
  3. Line 154, Table 3. I am not a statistician but the statistics in table 3 look off.  For total Fe doses, despite a moderate difference between the groups (favoring drug), there is little difference in the p values (P, 0.23 and drug, 0.15). In contrast, for number of patients receiving Fe, a decrease from 4 to 3 in placebo was p=0.8, while a decrease from 5 to 3 in drug was p=0.022.  How could a one patient difference between the groups lead to so large a p difference?
  4. Nasal endoscopy appearance is subject to considerable influence due to scope position/angle/depth and lighting. When we post a single image there is a natural tendency to select the images that best support our hypothesis. If a supplement is available for this article, it would be optimal for the authors to post before and after pictures for all or most patients from both groups.

Author Response

Our responses to the comments: 

Minor comments

  1. There is some inconsistency in the use of telangiectasia, telangiectasis, and telangiectases -thank you – was corrected.
  2. Line 70. Do we know anything about the timeline of response with propranolol gel and whether 8 weeks of treatment was optimal to detect a response? No. We did have a telephone call at 4 weeks and a subjective impression was recorded but we could not determine the exact timing. In future study we should look for an outcome to assess the time to response.
  3. Line 87. How was the determination of change made re endoscopy findings?  Were photos or videos taken at each timepoint and then compared?  Was it based on memory of the last endoscopy?  Based on notes? Thank you – we added to the Methods: LINE 105: "Patients’ nasal cavities were decongested with lidocaine 1.5% and phenylephrine 1% spray prior to endoscopic examination with a zero-degree 4mm endoscope connected to a high definition camera and monitor (Storz). Endoscopies were recorded and representative photographs of the nasal cavities were captured in a de-identified manner. Thus, all patient images were subsequently graded in an anonymized fashion at the conclusion of the study".
  4. Line 132. I do not understand the line below hemoglobin in table 2. The same numbers repeat in each column. We apologize- it was a tipo that was corrected
  5. Line 154. Epistaxis has been consistently shown to be the biggest determinant of QOL in HHT patients. Can the authors please discuss why placebo and drug patients showed a similar improvement in QOL despite a significant improvement in epistaxis being seen only in drug patients? After reviewing the questionnaires we think that SF 16 was probably not the best questionnaire for epistaxis QOL. This issue is discussed in the discussion- LINE 407- "QOL improved in both groups. The improvement in the placebo group may be related to several factors. First, a major component of the SF-16 questionnaire relates to the emotional state of the patient and its influence on his/her general health. The close follow up and care, the hope for improvement when joining the open-label phase, as well as the moistening effect of the gel, might have contributed to improvement in the patients' well-being and in QOL.  A questionnaire that is more specific to the effects of epistaxis might have shown a statistically significant difference between the groups".
  6. Line 164, Figure 2. Cell “A” is mentioned in the legend but not in the figure- figure was deleted as per the reviewers' comments.
  7. Please comment on whether the burning sensation may have been a subconscious clue to patients that they were receiving active drug and could have affected perceived response. Was there any correlation between burning and effect on epistaxis? This is a very important issue- we could not know whether this was considered. When discussed with patients we were very clear that it can be as a response to the preservatives (as we thought also). Since we are not aware of such side effect in topical propranolol in hemangiomas we were also puzzled by this side effect.

Major comments:

  1. Line 61. Use of antiangiogenics in the past 1 month is listed as exclusionary. Use of systemic pazopanib or bevacizumab in the past 3 months could potentially influence results due to known prolonged time course of response.  Did any patients receive either of these in the prior 3 months? Thank you for this comment. No patient was treated with any antiangiogenic treatment at all. In future studies this exclusion criteria should be addressed and corrected.
  2. Line 125. It is interesting that at study end there were 3 more placebo patients recruited relative to drug (13 vs 10), and that 3 placebo dropped out to give a final count of 10 in each group.  Did the randomization process somehow track this and attempt to maintain equal final distribution of the groups? Yes. We reported the pharmacy and only one pharmacist was unblinded and randomized accordingly.
  3. Line 154, Table 3. I am not a statistician but the statistics in table 3 look off.  For total Fe doses, despite a moderate difference between the groups (favoring drug), there is little difference in the p values (P, 0.23 and drug, 0.15). In contrast, for number of patients receiving Fe, a decrease from 4 to 3 in placebo was p=0.8, while a decrease from 5 to 3 in drug was p=0.022.  How could a one patient difference between the groups lead to so large a p difference? Thank you – we were asked to change the presentation of the secondary outcomes so the former table 3 was omitted and a new table with new statistics is presented as table 3 in line 237. 
  4. Nasal endoscopy appearance is subject to considerable influence due to scope position/angle/depth and lighting. When we post a single image there is a natural tendency to select the images that best support our hypothesis. If a supplement is available for this article, it would be optimal for the authors to post before and after pictures for all or most patients from both groups.We can prepare a file with all pictures and- if possible, it can be added as a supplement